# Association between Maternal Smoking during Pregnancy and Missing Teeth in Adolescents

**DOI:** 10.3390/ijerph16224536

**Published:** 2019-11-16

**Authors:** Junka Nakagawa Kang, Yuko Unnai Yasuda, Takuya Ogawa, Miri Sato, Zentaro Yamagata, Takeo Fujiwara, Keiji Moriyama

**Affiliations:** 1Department of Maxillofacial Orthognathics, Tokyo Medical and University, Tokyo 113-8510, Japan; junka14@yahoo.co.jp (J.N.K.); yuko.unnai.yasuda@gmail.com (Y.U.Y.); t-ogawa.mort@tmd.ac.jp (T.O.); 2Department of Health Sciences, Interdisciplinary Graduate School of Medicine and Engineering, University of Yamanashi, Yamanashi 409-3898, Japan; miris@yamanashi.ac.jp (M.S.); zenymgt@yamanashi.ac.jp (Z.Y.); 3Department of Global Health Promotion, Tokyo Medical and Dental University, Tokyo 113-8519, Japan

**Keywords:** cohort studies, Hypodontia, malocclusion, tobacco

## Abstract

Tooth agenesis and disturbance of tooth eruption is the most prevalent oral defect, and is possibly caused by the interaction of genetic and environmental factors. We hypothesized that prenatal factors may affect tooth development. The objective of this study was to examine whether smoking during pregnancy was associated with missing teeth in the offspring during adolescence. The study population comprised pregnant women and their children registered (*N* = 1052) at Koshu city, Japan. When the expectant mothers visited the city office for pregnancy registration, a survey was conducted to ascertain their lifestyle habits. Data on missing teeth in the children were obtained from the compulsory dental health checkup during junior high school years. Multivariate logistic regression models were fitted to assess the association between missing teeth and lifestyle habits. A total of 772 children were studied. The prevalence of missing teeth in these children was 4.9%. Children whose mothers smoked six cigarettes or more per day were 4.59 (95% CI: 1.07–19.67) times more likely to present with missing teeth than those children whose mothers did not smoke, after adjustment for possible confounders. Our findings indicate that smoking during pregnancy can be a risk factor for missing teeth in the offspring.

## 1. Introduction

Malocclusion, which is a developmental disorder of the jaws and teeth, can play an important role in both social interactions and functions [1]. A disproportionate number of teeth, which is one type of malocclusion, induces an unharmonious arrangement, such as crowding or spacing, in the dental arch, which clinicians must consider when making orthodontic and other treatment plans [2]. It also affects appearance, which may affect an individual’s self-esteem, communication, and quality of life [3,4]. Patients with missing teeth may suffer from further complications, such as periodontal damage, lack of alveolar bone growth, impaired chewing ability, and inaccurate pronunciation [5,6]. Malocclusion has also been associated with common physical symptoms, such as headache [7], and poor academic performance among adolescents [8].

The etiology of missing teeth, that is, a disturbance of tooth development (i.e., tooth agenesis or disturbance of tooth eruption), especially of the permanent teeth, remains unknown. The etiology of tooth agenesis includes genetic factors, environmental factors, or a combination thereof [9,10,11,12], while the etiology of a disturbance of tooth eruption may include systemic factors, local factors, as well as genetic factors [13,14,15,16]. For example, prolonged retention of a deciduous tooth due to failure of deciduous tooth resorption, abnormal an eruptive path, the presence of a supernumerary tooth, or anomalous position of a tooth (e.g., tilting, displacement, transmigration), can disturb the eruption of permanent teeth.

In human fetuses, deciduous tooth development starts around the sixth week of gestation as a morphologically distinct thickening of the oral ectoderm, called the dental lamina, and the first permanent tooth germs are observed between the 10th and 13th weeks of gestation [17]. Permanent tooth development involves tooth formation and eruption. All of the permanent teeth, except for the third molars, typically have erupted by age 13; thus, exposure to unfavorable factors during the critical period may disturb tooth development [18]. Tooth development depends on a series of inductive events involving genes coding for growth factors, such as those of the FGF, BMP, Wnt, and Hedgehog families, which regulate epithelial–mesenchymal interactions [19], and whose function can be regulated by environmental factors, such as anticancer drugs or antibiotics [20] or infections during pregnancy (such as rubella) [21], and maternal smoking during pregnancy [22].

Among several possible environmental risk factors for disrupted tooth development, maternal smoking during pregnancy warrants investigation, because previous studies have revealed a robust association between smoking during pregnancy and cleft lip and palate [23]. Thus, we hypothesized that maternal smoking during pregnancy would have adverse effects on tooth development, such as missing teeth, in the offspring. A previous study showed an association between maternal smoking during pregnancy and their children’s tooth agenesis; however, that study was a retrospective, hospital-based, case-control study, which may have been prone to recall bias [22].

To address the shortcomings of the previous study, in this prospective longitudinal, population-based study, we examined environmental factors, particularly during early pregnancy, that were associated with missing teeth.

## 2. Materials and Methods

### 2.1. Participants and Study Design

In Japan, pregnant women must register at a city office, and after delivery, the children must be registered by their parents. The study population recruited pregnant women who registered at the city office in Koshu city, Yamanashi prefecture, Japan, between 2 April 1996, and 1 April 1999, and between 2 April 2000, and 1 April 2003, as well as their children. The subjects were participants of Project Koshu (formely Project Enzan), a dynamic, ongoing prospective cohort study consisting of pregnant women and their children from a rural area in Japan, which commenced in 1988. Koshu city has a population of 27,000, with about 200 births per year. We expected a high-follow-up rate in this project because most of the population in this city have not migrated elsewhere, and used part of the data obtained from this project in the current study. 

For this study, when pregnant women registered at the city office, they were invited to complete a questionnaire survey to ascertain their lifestyle habits, after obtaining informed consent. Over 95% of expectant mothers in Koshu city registered before 16 weeks of pregnancy. The children were followed from birth onwards. Next, we obtained data regarding the gestational age at birth and the birth weight, which had been recorded in the Maternal and Child health handbook (boshi-techo) by the obstetrician or midwife in charge of delivery, when the children underwent a medical checkup at school at the age of 12–15 years. In this study, we approached all students (*N* = 1342) enrolled in three junior high schools in Koshu city (formely Enzan city), except for the students who were absent on the day of the dental examination. The corresponding data from the pregnancy period were prospectively followed (*N* = 772, follow-up rate: 73.3%). 

This study was approved by the Ethical Review Board of Tokyo Medical and Dental University (No. 1171) and the University of Yamanashi, School of Medicine (No. 332). We followed STROBE guidelines.

### 2.2. Smoking during Pregnancy

At their first pregnancy checkup, pregnant mothers were asked about their smoking status during pregnancy using a self-reported questionnaire that included questions about their smoking status before and during pregnancy (current or not) and the number of cigarettes they smoked per day. Those mothers who had smoked before but not during pregnancy were classified as ex-smokers. Those who smoked during pregnancy were classified as current smokers, and sub-grouped according to the number of cigarettes smoked daily: 1–5 and ≥6 cigarettes per day. Participants with missing information about maternal smoking status during pregnancy were excluded from the analyses.

### 2.3. Assessment of Missing Teeth

Missing teeth were assessed by three orthodontists from the Department of Maxillofacial Orthognathics of Tokyo Medical and Dental University, as a part of compulsory dental health checkups at junior high schools in Koshu city. Participants were examined under ambient lighting while seated in chairs. Orthodontic treatment history and extracted history due to caries or other reasons was ascertained by interviewing the participants. Missing teeth were diagnosed when one or more permanent teeth, other than third molars, were missing. The outcome variable was binary (having one or more or having no missing teeth).

### 2.4. Covariates

Maternal lifestyle during pregnancy was considered on the basis of a life course perspective about the development of oral disease [24]. Thus, the following possible covariates were obtained from the questionnaire: Sex of the child, gestational age (full term, <37 weeks), maternal body mass index (BMI) before pregnancy (normal weight, overweight: 25–30 kg/m^2^, obesity: ≥30 kg/m^2^+), maternal age at delivery (<20, 21–30, 31–40, >40 years), maternal education (junior high school or high school, some college education, college or more), alcohol consumption during pregnancy (yes, no), paternal smoking during maternal pregnancy (yes, no), and maternal breakfast consumption habits (every day, 3–5 times a week, 1–2 times a week, none). 

### 2.5. Statistical Analysis

The associations between missing teeth and possible risk factors were analyzed using multiple logistic regression. Maternal smoking habit status was categorized into three levels (never and stopped smoking before pregnancy, sustained smoking during pregnancy 1–5, or smoking ≥ 6 cigarettes/day). The regression model was adjusted for covariates, including the child’s sex and gestational age, and variables showing significant association with maternal smoking, except for maternal drinking, due to the limited sample size. The significance level was set at *p* < 0.05 (two-sided). All analyses were conducted using the Stata/SE 14.0 software package (STATA Corporation, College Station, TX, USA).

## 3. Results

A total of 772 children were studied. Table 1 shows the distributions of sociodemographic characteristics, sex, grade, birth weight, gestational age, delivery, maternal BMI, maternal age at delivery, parental education, and maternal exposures during pregnancy stratified by maternal smoking during pregnancy. The prevalence of missing teeth in these children was 4.9%. The percentage of mothers who smoked during early pregnancy was 6.0%. Regarding covariates, a higher number of younger mothers than older mothers were observed among participants who smoked during pregnancy. Lower levels of parental education were associated with increased maternal smoking during pregnancy. In general, mothers who smoked during pregnancy were more likely to have spouses who smoked heavily, to drink alcohol during pregnancy, and to skip breakfast. Sex, birth weight, gestational age, and maternal BMI were not associated with maternal smoking status during pregnancy, as determined via the Chi-square test. 

Figure 1 shows selection of the participants for the study. The adjusted odds ratios (ORs) and 95% confidence intervals (CIs) for the maternal factors that were associated with missing teeth are presented in Table 2. According to the crude model, maternal smoking of 6 or more cigarettes per day during early pregnancy was associated with missing teeth in their offspring (OR: 3.47, CI: 1.14–10.56), relative to mothers who did not smoke during pregnancy. After adjusting for covariates, maternal smoking of 6 or more per day cigarettes during early pregnancy remained significantly associated (OR: 4.59, CI: 1.07–19.67). Furthermore, although non-significant, mothers who smoked 1–5 cigarettes per day during pregnancy also showed a tendency for missing teeth in the offspring (OR 2.80, CI: 0.52–15.06), suggesting a dose-response association between maternal smoking during pregnancy and missing teeth in the offspring (*p* for trend = 0.024). Figure 2 shows the adjusted difference (with 95% confidence intervals) according to maternal smoking categories (6 or more per day cigarettes) during early pregnancy.

## 4. Discussion

In this study, we found an association between perinatal environmental factors, such as exposure to maternal smoking, and missing teeth. The association remained statistically significant even after adjusting for potential confounding variables. Additionally, as the demonstration of a dose-response effect is an important and powerful proof of causation of a condition by an exposure, our finding of a statistically significant dose-response trend with increased odds of missing teeth with increased maternal cigarette smoking during pregnancy is important.

This study adds to the body of knowledge by showing the association between maternal smoking and missing teeth in the offspring using a population-based sample. Al-Ani et al. showed that maternal smoking during pregnancy is associated with offspring tooth agenesis using a hospital-based sample [22]. That study was based on self-reported exposure; therefore, the data could have been influenced by recall bias. We performed a retrospective cohort study, in which we linked maternal perinatal data with dental checkup data of the children in junior high school, to clarify the association between maternal smoking during pregnancy and having a child with missing teeth. Such an association in a birth cohort study has not been reported previously.

The detailed etiology of missing teeth remains unknown, although it is well-established that it involves a combination of genetic and environmental factors. Exposure of children to tobacco components during fetal development and exposure to environmental tobacco smoke is presumed be the most common and dangerous environmental exposure experienced by children, and hence it has attracted considerable attention [25,26]. Toxic tobacco compounds might alter the expression of genes involved in the development of teeth and cranial bone due to changes in DNA methylation, which have been observed in women who smoke during pregnancy [27]. Several studies have investigated whether maternal smoking increased the risk of having offspring with cleft lip and palate [23,28,29]. Additionally, both carbon monoxide and nicotine exposure, commonly produced by cigarette smoking, can produce tissue hypoxia [30].

Specific developmental cascades are common to the morphogenesis of both teeth and some craniofacial structures. Maxillofacial structures, such as bone, nerve, and connective tissues, are all generated from neural crest cells [18]. These cells undergo epithelial to mesenchymal transformation and migrate to various locations in the body where they contribute to the formation of a wide variety of tissues. The mechanism by which maternal smoking during early pregnancy may underlie the offspring’s missing teeth may involve the following. First, neural crest cells are exposed to oxidative stress, caused by smoking, during the tooth developmental stage. Neural crest cells are very sensitive to oxidative stress [31], and it is accepted that smoking during pregnancy increases levels of oxidative stress [32]. It is possible that the additional carbon dioxide caused by the poor oxygenation involved with smoking may lead to acidosis, which in turn could disturb tooth development. Other studies have concluded that increased cigarette smoking during pregnancy resulted in increased odds of having a child with cleft lip and palate [23]. Oxidative stress by maternal smoking may thus prevent proper development of neural crest cell-derived deciduous teeth and permanent teeth; consequently, it is reasonable to suppose that oxidative stress, due to maternal smoking during pregnancy, contributes to missing teeth in the offspring.

Second, nicotine is considered to be a key teratogenic substance that alters and delays embryonic development [33]. Depending on the amount of nicotine absorbed during smoking, contraction of the arterioles, decrease in blood flow velocity in the capillary blood vessels, and inhibition of supply of oxygen and nutrients to the surrounding tissues could lead to tissue injury [34]. Nicotine exposure during pregnancy is reported to increase the risk for oral cleft lip and palate. Additionally, nicotine ingested by the mother can pass through the placenta and affect the fetus in mice [34]. Maternal smoking during pregnancy restricts the blood flow in the vascular beds of the placenta [35]. The oxygen deficiency in the placental circulation can result in chronic fetal hypoxia and subsequent fetal growth retardation, which may hinder proper development of teeth during the development stage [36]. Nicotine may also interfere with reciprocal induction between ectomesenchymally-derived tissues and oral ectoderm, disrupting subsequent normal tooth development. Furthermore, at the cellular level, nicotine impairs angiogenesis and the proliferation of erythrocytes; fibroblast proliferation and adhesion, collagen synthesis, and osteogenesis [37], while it also induces osteoblastic apoptosis and increases osteoclastic activity [38]. Considering these effects, it is reasonable to suppose that nicotine affects resorption of the deciduous teeth or normal eruption of the permanent teeth.

The current study had several strengths. First, we used a population-based prospective dataset and had a high follow-up rate; thus, we can generalize the evidence at least to the population of Koshu city in Japan. There has been no other nationwide community-based study examining the effect of smoking status during pregnancy on missing teeth in the offspring, while simultaneously controlling for confounding factors, which may have influenced the results of previous studies. We used a self-reported questionnaire at the time when mothers realized that they were pregnant; thus, the study was not subject to recall bias. It is likely that we minimized the biases of exposure measurements. Therefore, this study provided valuable evidence supporting the importance of cessation of maternal smoking before and during pregnancy.

However, our study was not free of limitations. First, this study assessed smoking by self-report, and therefore socially desirable bias, that is, underestimation, is possible. Nevertheless, recent studies in other countries showed that self-reported smoking status was a valid marker for tobacco exposure [39]. Second, we did not obtain the site of missing teeth; nevertheless, recent studies showed that most pregnant women who smoked at week 16 of gestation continued to smoke throughout pregnancy [40]; thus, these adverse effects were likely to have been present throughout tooth development from the initiation of tooth eruption. Third, there may be other unmeasured or residual confounding factors, such as diet during pregnancy, other lifestyle factors involving various settings, or time of smoking cessation and change in smoking habits, that cannot be ruled out. Fourth, we did not measure maternal smoking status objectively. Further studies incorporating standardized assessment, such as the Fagerstrom test, are needed. Another important limitation is the low number of cases with missing teeth, which led to wide confidence intervals, suggesting lower precision and wider variance of the results.

## 5. Conclusions

In conclusion, we found a statistically significant risk of missing teeth in the offspring whose mothers smoked during pregnancy. Our findings provide further support for the current recommendation of abstinence from smoking during pregnancy, based on the notion that there is no known safe threshold of maternal smoking during pregnancy. Future studies should seek insights into the biological effect of maternal smoking on tooth development and the etiology of missing teeth.

## Figures and Tables

**Figure 1 ijerph-16-04536-f001:**
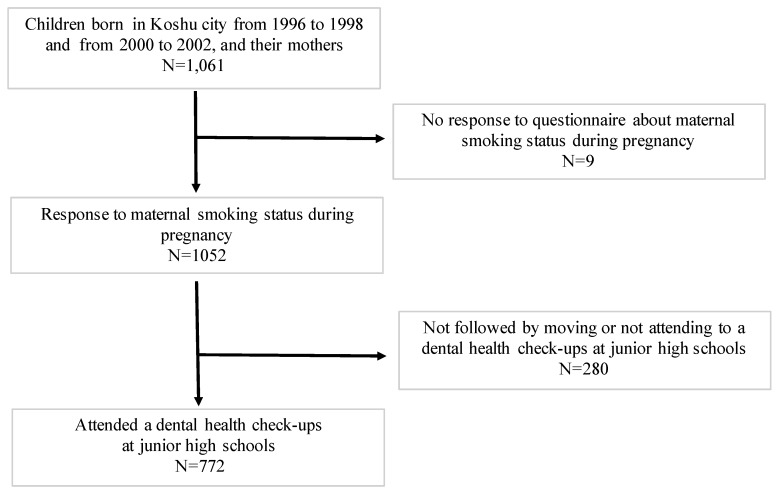
Case selection flow chart.

**Figure 2 ijerph-16-04536-f002:**
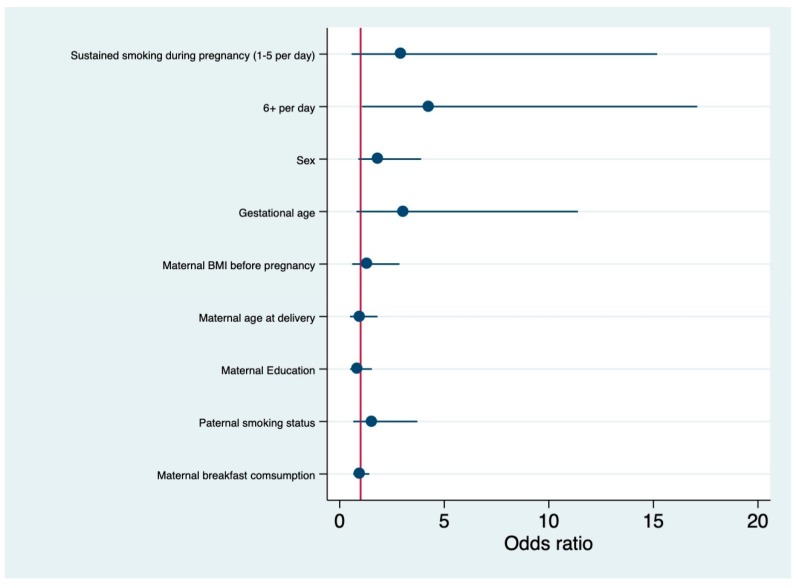
Adjusted odds ratios for missing teeth.

**Table 1 ijerph-16-04536-t001:** Comparison of sociodemographic characteristics between those with missing teeth and those with non-missing teeth.

Characteristics	Missing Teeth (−)		Missing Teeth (+)		*p* for Chi-Square Test
	(*n* = 734: 95.1%)		(*n* = 38: 4.9%)		
	*n*	%	*n*	%	
Sex					
Male	400	54.5	19	50.0	0.59
Female	334	45.5	19	50.0	
Grade					
1	228	31.1	8	21.1	0.42
2	265	36.1	16	42.1	
3	241	32.8	14	36.8	
Time of registration of pregnancy					
Early (<16 gestational weeks)	623	94.3	30	88.2	0.15
Late (16 gestational weeks+)	38	5.7	4	11.8	
Birth Weight					
Normal (2500 g+)	673	93.1	33	91.7	0.75
Low (<2500 g)	50	6.9	3	8.3	
Gestational age					
Full term (37 weeks+)	689	95.4	33	91.7	0.30
Preterm (<37 weeks)	33	4.6	3	8.3	
Delivery					
Normal	539	80	25	73.5	0.19
Caesarean	111	16.5	9	26.5	
Suction	24	3.5	0	0.0	
Maternal age at delivery					
<20 years	6	0.8	1	2.6	0.57
21–30 years	362	49.7	19	50.0	
31–40 years	346	47.6	18	47.4	
40+ years	14	1.9	0	0.0	
Duration of Exclusive breastfeeding					
Never	45	6.3	2	5.7	0.42
<6.0 months	547	76.8	30	85.7	
6.0 months+	120	16.9	3	8.6	
Maternal Education					
JHS or HS	288	44.7	16	47.1	0.42
Some college	268	41.6	16	47.1	
College or more	88	13.7	2	5.8	
Paternal Education					
JHS or HS	312	48.6	21	61.8	0.32
Some college	93	14.5	4	11.8	
College or more	237	36.9	9	26.4	

JHS, junior high school; HS, high school.

**Table 2 ijerph-16-04536-t002:** Unadjusted and adjusted odds ratios for missing teeth.

Variables	Number of Non-Missing Teeth	Number of Missing Teeth	Crude		Adjusted	
			OR	95% CI	OR	95% CI
Maternal smoking status						
Never or stopped smokers before pregnancy	694	32	Ref		Ref	
Sustained smoking during pregnancy (1–5 per day)	15	2	2.89	0.63–13.19	2.80	0.52–15.06
(6+ per day)	25	4	**3.47**	**1.14–10.56**	**4.59**	**1.07–19.67**
*p*-value			**0.014**		**0.024**	
Sex						
Male	400	19			Ref	
Female	334	19			1.92	0.89–4.14
Gestational age						
Full term (37 weeks+)	689	33			Ref	
Preterm (<37 weeks)	33	3			3.82	0.96–15.15
Maternal BMI before pregnancy						
Normal weight (<25.0 kg/m^2^)	624	29			Ref	
Overweight (25.0–30.0 kg/m^2^)	52	4			1.84	0.58–5.80
Obesity (30.0 kg/m^2^+)	19	1			0.74	0.07–8.01
Maternal age at delivery						
<20 years	6	1			1.87	0.16–21.45
21–30 years	362	19			Ref	
31–40 years	346	18			1.10	0.52–2.33
40+ years	14	0			NA	
Maternal Education						
JHS or HS	288	16			Ref	
Some college	268	16			1.19	0.54–2.61
College or more	88	2			0.48	0.10–2.25
Maternal alcohol consumption						
(−)	639	37			Ref	
(+)	87	1			NA	
Paternal smoking status during maternal pregnancy						
(−)	187	6			Ref	
(+)	46	2			1.57	0.64–3.84
Maternal breakfast consumption habits						
Every day	569	29			Ref	
3–5 times a week	51	1			0.36	0.04–2.95
1–2 times a week	49	5			1.54	0.49–4.79
None	63	3			0.61	0.13–2.85

Bold signifies *p* < 0.05.

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
