# Peer review of "Association between Maternal Smoking during Pregnancy and Missing Teeth in Adolescents"

_ijerph, 2019, doi:10.3390/ijerph16224536_

Round 1

Reviewer 1 Report

I am grateful for the opportunity to review the manuscript presented to me. I hope that the comments in the review would be helpful in deciding whether to publish the manuscript in your journal. I believe the paper is worth considering for publication, however requires minor revision. All information you will find in the attached text.

Reviewer 2 Report

The manuscript presented for the review deals with a very important issue which is lifestyle of pregnant women, including habits like smoking, that may affect not only developing foetus, but also health-status of children in their future life. Since not many studies concerned the topic presented in the manuscript, I believe that this is a valuable contribution to the science and general understanging of the risky behaviors during pregnancy effect on the future life of progeny.

minor comments:

line 34 - is "relationship" a good word here? Maybe "arrangement" or something like that would be better?
line 67 - the "word" "here" can be removed
Material and methods section:
I assume that the study included identified pairs mother-child, am I right?
The authors stated that the examined women were registered in the years 1996-1999, and 2000-2003. What was then the age of children examined? Is it possible that the differentiated age of children affected the results of the study?
I also feel that it would be reasonable to mention about the causes of missing teeth in children examined.
line 192 - the word "is" at the and of the line should be removed
